# Role of Lutetium Radioligand Therapy in Prostate Cancer

**DOI:** 10.3390/cancers16132433

**Published:** 2024-07-01

**Authors:** Ignacy Książek, Artur Ligęza, Franciszek Drzymała, Adam Borek, Marcin Miszczyk, Marcin Radosław Francuz, Akihiro Matsukawa, Takafumi Yanagisawa, Tamás Fazekas, Łukasz Zapała, Paweł Rajwa

**Affiliations:** 1Department of Urology, Medical University of Silesia, 41-800 Zabrze, Poland; ignacyk98@gmail.com (I.K.); adam_b1998@interia.pl (A.B.);; 2Department of Urology, Medical University of Vienna, 1090 Vienna, Austria; 3Collegium Medicum—Faculty of Medicine, WSB University, 41-300 Dąbrowa Górnicza, Poland; 4Department of Urology, Jikei University School of Medicine, Tokyo 105-8461, Japan; 5Department of Urology, Semmelweis University, 1085 Budapest, Hungary; 6Centre for Translational Medicine, Semmelweis University, 1085 Budapest, Hungary; 7Clinic of General, Oncological and Functional Urology, Medical University of Warsaw, 02-091 Warsaw, Poland

**Keywords:** prostate cancer, Lutetium-177-PSMA, radioactive isotopes, radiation therapy

## Abstract

**Simple Summary:**

Lutetium-177-PSMA—a radiopharmaceutical composed of a radionuclide lutetium-177 and a PSMA-binding ligand—is a theranostic medicine for prostate cancer. In this review, a summary of the impact of this drug on castration-resistant prostate cancer, hormone-sensitive prostate cancer, and localized prostate cancer is provided. Available data demonstrate that ^177^Lu-PSMA could be a successful therapy in those types of prostate cancers, but it is not without adverse effects. Conclusions revealed the need for further well-designed high-quality controlled trials to evaluate long-term outcomes and effectiveness.

**Abstract:**

Theranostics utilize ligands that chelate radionuclides and selectively bind with cancer-specific membrane antigens. In the case of prostate cancer (PCa), the state-of-the-art lutetium-177-PSMA combines the radioactive β-emitter ^177^Lu with Vipivotide Tetraxetan, a prostate-specific membrane antigen (PSMA)-binding ligand. Several studies have been conducted, and the therapy is not without adverse effects (e.g., xerostomia, nausea, and fatigue); however, few events are reported as severe. The available evidence supports the use of ^177^Lu-PSMA in selected metastatic castration-resistant prostate cancer patients, and the treatment is considered a standard of care in several clinical scenarios. Emerging research shows promising results in the setting of hormone-sensitive prostate cancer; however, evidence from high-quality controlled trials is still missing. In this review, we discuss the available evidence for the application of ^177^Lu-PSMA in the management of PCa patients.

## 1. Introduction

The treatment of prostate cancer (PC) has evolved significantly over the years—surgical techniques, chemotherapy, abundant forms of radiotherapy, androgen receptor signaling inhibitors (ARSIs), agonists and antagonists of luteinizing hormone-releasing hormone (LHRH), and poly (ADP-ribose) polymerase inhibitors (PARPi) represent the treatment landscape of PC. However, there are many effects associated with these treatments [1,2,3]. Despite a favorable prognosis in the localized and regional stages, patients with distant cancer still have a poor prognosis [4,5]. Therefore, novel agents are being investigated to elevate both the life expectancy and quality of life (QoL) of patients with advanced PC. One promising group of anti-cancer agents is based on theranostics, which describes a technique that, depending on the radionuclide and ligand, can be used for both therapeutic and diagnostic purposes. Molecularly-targeted radioligand therapies include the alpha and beta-emitters actinium-225 and lutetium-177.

Lutetium-177-PSMA is a radiopharmaceutical composed of a radionuclide lutetium-177 attached to a transmembrane glycoprotein known as prostate-specific membrane antigen (PSMA), also referred to as glutamate carboxypeptidase II or folate hydrolase. Conceptually, this should lead to highly selective delivery of radiation to the metastatic lesions, similar to radiotherapy-based metastasis-directed therapy (MDT) [6,7,8]. Lutetium-177 is an unstable isotope ($T_{1/2}$ = 6.65 days) produced in a particle accelerator called a cyclotron through the irradiation of ^176^Lu or ^176^Yb. The use of ^176^Lu is considered the direct route, as its primary product is ^177^Lu. Conversely, the utilization of ^176^Yb is termed the indirect route due to the short-lived intermediate ^177^Yb. Vipivotide tetraxetan chelates the radionuclide ^177^Lu and binds to human prostate-specific membrane antigens. Once administered intravenously, ^177^Lu-PSMA is distributed to targeted tissues within 2.5 h. There are no identified enzymes or transporters that interact with vipivotide tetraxetan ^177^Lu. Approximately 60–70% of ^177^Lu vipivotide tetraxetan binds to human plasma proteins and is primarily eliminated through the kidneys in its unchanged form. A decrease in creatinine clearance may extend exposure time, though the drug itself does not appear to be nephrotoxic. However, in the VISION trial, several cases of renal disorders were observed [9]. A portion of the drug can bind to the salivary glands, leading to a dry mouth. Higher tumor burden correlates with lower doses observed in other organs [10,11,12]. Post-administration, ^177^Lu decays to stable hafnium-177, emitting beta particles with a maximum energy of 0.497 MeV and maximal tissue penetration of 2 mm. This helps minimize damage to non-tumor tissues [13,14,15]. Moreover, gamma particles are emitted too, which is used in prostate cancer imaging [16].

The most studied theranostic PSMA ligands (PSMA-617 and PSMA Imaging and Therapy) are appropriate for therapeutic usage and have similar in vivo behavior, toxicity profiles, and efficacy in metastatic castration-resistant prostate cancer (mCRPC) patients [17]. Unlike normal tissues, prostate cancer cells express PSMA at levels substantially higher than those found in the kidney, gut, and salivary gland tissues. Therefore, these agents accumulate specifically in prostate cancer cells, causing DNA damage and signaling cell death while sparing healthy tissue [17,18]. Emerging prospective data suggests these agents’ efficacy is similar to that of cabazitaxel and beneficial for heavily pretreated patients with mCRPC [9,19]. In numerous early-phase investigations, radioligand therapy exhibits encouraging biochemical and radiographic response rates, reduced pain, and tolerable toxicity [9]. These studies are summarized in Table 1 and Table 2. The objective of this narrative review is to evaluate the existing evidence on the use of ^177^Lu-PSMA radioligand therapy in managing PC.

## 2. mCRPC

The mCRPC represents an advanced stage of PC with a notably poor prognosis due to resistance to androgen deprivation therapy (ADT). In this terminal disease phase, the treatment often comprises docetaxel, abiraterone/prednisolone (AAP), enzalutamide, cabazitaxel, olaparib, niraparib/AAP, and talazoparib/enzalutamide [30]. However, these therapeutic strategies often come with numerous adverse effects and limited efficacy. Patients with mCRPC are first-choice candidates for enrollment in clinical trials that evaluate novel, promising molecules for PC treatment. Notable studies conducted on a large patient cohort include the Phase 3 VISION trial by Sartor et al. [9]. However, despite the high number of patients included, the study provides a limited understanding of the role of ^177^Lu due to sub-standard treatment methods in the control arm and highly unbalanced dropout rates between arms [31]. It is possible that a considerable number of patients discontinue participation due to the availability of more efficient standard-of-care (SoC) therapeutic options than those allowed in the control arm, such as taxane-based chemotherapy.

Noteworthy is that ^177^Lu-PSMA efficacy was evaluated in heavily pretreated mCRPC patients and compared against a few SoC therapies [9,19]. Several moderators of ^177^Lu-PSMA-617 efficacy were identified, including the clinical stage at ^177^Lu-PSMA-617 treatment initiation, prior taxane-based chemotherapy, or prostate-specific antigen (PSA) decline following drug administration. In two studies, ^177^Lu-PSMA was compared to chemotherapy in chemotherapy-naive patients [20,21], and two studies compared ^177^Lu-PSMA against chemotherapy, suggesting non-inferiority of the treatment [19,20]. However, both studies used intermediate clinical endpoints, such as progression-free survival (PFS) and PSA response, that were recently shown to be insufficient for overall survival (OS) surrogacy, and survival outcomes need to be analyzed to verify the non-inferiority of the treatment [32,33].

In the TheraP study, there was no statistically significant difference in PFS between patients treated with ^177^Lu-PSMA and cabazitaxel [19]. Sartor et al. demonstrated an improvement in both OS (HR 0.62) (95% CI; 0.52 to 0.74; *p* < 0,001) and PFS (HR 0.4) (99.2% CI; 0.29 to 0.57; *p* < 0.001) in the study group. However, it is important to note that the trial included restrictive criteria for SoC therapy in the control group, which led to many patients being offered possibly sub-standard therapy, and unbalanced dropout rates between arms [9]. For example, the median PFS duration varied depending on whether the patient had prior taxane treatment (6 vs. 8.8 months; pre-treated vs. taxane-naïve) [21]. In a retrospective analysis by Barber et al., the median overall survival was higher in taxane-naive patients, lasting up to 27.1 months [21]. Sartor et al. prove that patients who underwent SoC and chemotherapy experienced a median overall survival between 11.3 and 15.3 months [9].

Although the objective responses achieved in this single-center prospective clinical trial showed promising results with minimal toxicity, it is important to note that the study had certain limitations. There were no specific initial criteria for selecting patients, and the patients were a diverse group. The individuals seeking medical treatment varied widely. PSMA RLT was considered a logical therapeutic choice, and the decision to pursue this treatment was made by oncologists and urologists after exploring all other available options [22]. It is recommended to implement lutetium treatment and include it in clinical trials at earlier disease stages, using the same criteria for inclusion and exclusion as in other studies, to facilitate comparison [9,19,20,21,22,23,24,25].

### 2.1. Long-Term Response

Several authors reported that reaching a given PSA reduction threshold is an indicator of a long-term response. An association between longer median PFS and PSA declines of over 50% was found by Hoffman et al. [23]. While ^177^Lu-PSMA appeared to induce a greater PSA decline than Cabazitaxel and Ra-223 [19,23], Violet et al. found that patients with a PSA decline of over 50% exhibited significantly longer median overall survival compared to those with less than 50% (18.4 months vs. 8.7 months) [24]. Rahbar et al. suggested that a PSA decline of approximately 20% is the most significant prognostic predictor for longer survival (HR 0.28) [25]. While these findings show that the concept of biochemical response is a promising predictor of survival in patients treated with ^177^Lu, it must also be noted that a posteriori analyses introduce a significant immortal-time bias and that biochemical endpoints were shown to be insufficient surrogates of survival in both metastatic hormone-sensitive prostate cancer (mHSPC) and mCRPC patients [32,33].

Furthermore, attention should be drawn to PSMA-based radioligand treatment in homologous recombination repair (HRR)-deficient PC patients. HRR deficiency is a phenotype defined by a cell’s incapacity to use the HRR pathway to efficiently repair double-strand DNA breaks. Privé et al. reported that DNA damage repair defects are not significantly associated with exceptional responsiveness to PSMA therapy [33,34]. However, further studies in larger prospective cohorts are necessary to verify the association between DNA damage repair gene defects and differential responses to PSMA-RLT.

### 2.2. Quality of Life

When assessing treatment options, the impact on a patient’s QoL should be carefully considered, with special emphasis on pain relief, as it can significantly enhance a patient’s life quality. For individuals with mCRPC, it is crucial to focus on improving their QoL and minimizing harmful treatment side effects, considering that many are already burdened with disease symptoms. ^177^Lu-PSMA seems especially adept at relieving pain, with a considerable number of patients reporting rapid relief following the initial treatment cycle. Among 27 patients who had initial pain, 37% noted improvement after their first treatment round. This amount of pain reduction is promising and is comparable to that obtained with other medications, such as cabazitaxel. However, particular attention should be paid to the level of pain that patients enroll in the trial with, as this may affect the test results and interpretation of the efficacy of lutetium [23]. The therapy also showed greater improvements in fatigue, social functioning, and insomnia compared to cabazitaxel. Therefore, ^177^Lu-PSMA-617 may be appropriate for individuals who are ineligible for other forms of therapy owing to age or comorbidities [19]. Another study indicated that ^177^Lu-PSMA-617 had improved health-related QoL compared to docetaxel. More extensive studies should be conducted with larger groups of participants to assess how ^177^Lu-PSMA-617 performs against other interventions in the initial treatment of mCRPC [20]. According to all the studies conducted, the treatment has exhibited minimal toxicity and high tolerability, with rare occurrences of dose reduction or discontinuation of the medication [9,19,20,21,22,23,24,25].

### 2.3. Take Home Message

Further research is required to investigate correlations between the effectiveness of ^177^Lu-PSMA at different dosages, as well as to evaluate long-term outcomes in a larger patient cohort, building upon studies such as Satapathy et al. Currently, the available evidence suggests that ^177^Lu-PSMA is a successful therapy with tolerable toxicity rates for heavily pretreated mCRPC patients expressing PSMA who have progressed after other lines of therapy. The results of mCRPC research serve as a foundation for further studies in other patient groups, such as localized PCa and mHSPC [20,21,24,25].

## 3. Localized

The unequivocal role of neoadjuvant therapies, such as chemohormonal or androgen deprivation therapy, for unfavorable localized prostate cancer is yet to be determined. Despite the absence of concrete evidence demonstrating improvement in oncological outcomes from neoadjuvant therapy preceding radical prostatectomy [35,36], nearly 65% of patients require supplemental therapy post-surgery, and a third will ultimately experience progression [37,38]. In such cases, ^177^Lu-PSMA emerges as a promising agent due to its direct targeting of cancer cells, inclusive of micrometastases, potentially preventing disease progression.

Golan et al. conducted a clinical trial with 14 patients with high-risk localized prostate cancer [26]. Before robot-assisted radical prostatectomy (RARP), patients were treated with 2–3 doses of ^177^Lu-PSMA administered at two-week intervals. The median PSA reduction registered was 3.45 ng/mL (IQR 0.4–5.2) after two, and 4.3 ng/mL (IQR 0.92–6.8) after three ^177^Lu-PSMA doses. The salivary gland damage was reduced by applying ice packs 10 min before and 30 min post-administration. All prostatic specimens showed adenocarcinoma upon histological examination, with a positive surgical margin observed in seven patients (53%). Radiation-induced changes, such as clear vacuolated cytoplasm and pyknotic nuclei, were evident. The authors did not observe anatomical alterations in the prostate area, and no significant intraoperative complications were recorded. Most patients required one or fewer pads three months post-surgery, and 50% retained erections. Overall, the treatment was well-tolerated [26]. In the LuTectomy clinical trial, 20 patients received ^177^Lu-PSMA intravenously before a planned radical prostatectomy and pelvic lymph node dissection. Almost half of the patients achieved a 50% PSA reduction at 6 weeks after administrating ^177^Lu-PSMA, and the treatment was well tolerated. In conclusion, ^177^Lu-PSMA shows initial efficacy as a neoadjuvant treatment before radical prostatectomy, and the treatment appears to be safe and feasible. Certain estimates suggest that ^177^Lu-PSMA could not only augment surgery but even possibly replace it as the first line setting in selected patients in the future [27,39].

### Take Home Message

^177^Lu-PSMA appears to hold promise as a therapeutic option in the context of localized disease, in particular as a neo-adjuvant treatment. However, the preliminary results should be verified in prospective, controlled studies.

## 4. mHSPC

Currently, there is a wide range of approved agents for the treatment of mHSPC. Still, the backbone of the therapy involves combining ADT with ARSIs (abiraterone, apalutamide, or enzalutamide) and the option to add docetaxel. The treatment protocol also encompasses the combination of ADT, darolutamide, and docetaxel. Additionally, patients with low metastatic burden may benefit from prostate radiotherapy, cytoreductive prostatectomy, or MDT [8,40,41,42]. The selection of therapy should be individualized, taking into account various factors such as expected therapy results, the patient’s condition, disease burden, patient preferences, and treatment toxicity. Prive et al. conducted a prospective pilot study assessing the efficacy of ^177^Lu-PSMA in 10 patients with low-volume mHSPC [28]. Half of the patients experienced a PSA reduction of >50%, with one patient achieving an undetectable PSA. At 24 weeks after the second cycle, five patients displayed at least a partial response, with one patient revealing a complete response, and the disease progressed in four patients. The most common symptoms reported were fatigue that subsided after 2–4 weeks. Other complaints included nausea, xerostomia, rash, and pain. In laboratory tests, no signs of kidney, liver, or bone marrow damage were observed. Moderate side effects may result from using a low dose and low cycle number; hence, there is a need for further research to evaluate the safety of higher dose and cycle numbers. Currently, a third-phase clinical trial of PSMAddition is ongoing to assess the effectiveness and safety of ADT and ARSI in combination with ^177^Lu-PSMA-617 therapy compared to androgen deprivation therapy and ARSI alone. In this study, about 1126 patients will be randomized in a 1:1 ratio and receive up to 6 cycles of ^177^Lu-PSMA-617 (7.4 GBq) every 6 weeks [43]. BULLSEYE is an ongoing prospective randomized multicenter II phase study that compares ^177^Lu-PSMA-617 to the current standard of care in oligometastatic HSPC. It is also being investigated whether ^177^Lu-PSMA-617 can delay ADT [29].

### Take Home Message

Preliminary results suggest that ^177^Lu-PSMA is a promising therapy for mHSPC, but further investigation is necessary to confirm these conjectures.

## 5. Adverse Effects of Lutetium Therapy

Among the adverse effects of ^177^Lu-PSMA, mild and moderate ones were predominant, with only a few incidents reported as severe or life-threatening. A significant retrospective analysis at various centers showed that xerostomia, nausea, and fatigue were experienced by 8%, 6%, and 13% of patients, respectively. Grade 3–4 hematologic adverse events were recorded in just 12% of patients, where anemia was found to occur in 10% of instances [44]. As this research was retrospective, underreporting of adverse events is a common issue and might be the case here. Prospective trials are therefore more dependable for monitoring adverse events [45]. An international phase III open-label prospective study found that the most common complications were fatigue (43.1%), dry mouth (38.8%), and nausea (35.3%), most of which were Grade 1 or 2 according to the CTCAE (Common Terminology Criteria for Adverse Events) [9]. Concerns have been raised regarding disorders of the blood and lymphatic system, with anemia being the most common. Symptoms classified as Grade 3 or higher occurred in 12.9% of patients, and overall, 31.8% of patients had anemia. Careful consideration should also be given to other conditions such as thrombocytopenia (17.2%), lymphopenia (14.2%), and leukopenia (12.5%). These hematological complications are more likely to present as grade 3 or above than non-hematological complications [9]. However, information on potential long-term adverse effects, which may be of great importance, is missing from the present study. Despite the documented side effects, just a small percentage of patients (11.9%) withdrew from treatment or needed to lower their dosage (5.7%). These findings highlight the overall safety of this therapy [9]. While previous studies have mainly focused on the immediate post-treatment period, future research should investigate the long-term effects associated with the use of ^177^Lu-PSMA [46].

## 6. The Future of Lutetium Therapy

Beta emitters such as ^177^Lu have demonstrated therapeutic efficacy in treating large tumor masses due to their long radiation range and ability to induce a cross-fire effect [47,48]. Targeted alpha-radiation therapy (TAT) using ^225^Ac may be even more effective in treating patients with disseminated metastatic disease due to the shorter radiation range coupled with high-linear-energy transfer that induces targeted tumor cell killing by causing a higher number of double-strand DNA breaks in tumor cells, resulting in their destruction, while minimizing damage to surrounding tissues for instance red bone marrow. This treatment shows promise and requires further research with larger cohort studies. These studies should include ^177^Lu and ^225^Ac, as well as immunohistochemical analysis and genomic profiling of patients with mCRPC [49].

## 7. Conclusions

The treatment of prostate cancer is still evolving. Collected evidence supports lutetium-177-PSMA as a standard-of-care treatment for castration-resistant prostate cancer. Other research shows promising results in the management of hormone-sensitive prostate cancer and localized prostate cancer. However, therapy has its drawbacks, such as nausea, xerostomia, and fatigue. Despite optimistic results, further research is required with a larger cohort to assess long-term outcomes and efficiency with various patient groups.

## Figures and Tables

**Table 1 cancers-16-02433-t001:** The basic characteristics of the studies on the use of ^177^Lu-PSMA in the treatment of prostate cancer (original work).

Study	Phase	Type of Study	Origin (Country)	Patients Total S Arm (Study) C Arm (Control)	Median Age	Primary mCRPC or Pre-Treated	PSA Response	Recruitment Years	Follow up Period	Drugs Used
*mCRPC*										
VISION trial, Sartor et al. [9]	3	Open-label, randomized prospective	International	S 551 C 280	S 70.0 C 71.5	Pre-treated	S 71.5% C 35.5%	2018–2019	20.9 months	^177^Lu-PSMA-617
TheraP Hofman et al. [19]	2	Prospective randomized, open-label	Australia	S 99 C 101	S 72.1 C 71.8	Pre-treated	S 66% C 37%	2018–2019	Mean 18.4 months	S ^177^Lu-PSMA-617 C Cabazitaxel
Satapathy et al. [20]	2	Parallel group	India	S 20 C 20	68	no chemotherapy, only novel androgen-axis drugs	S 50% C 40%	2019–2021	20 months	^177^LuPSM-617, docetaxel
Barber et al. [21]	-	Retrospective	Australia	S1 83 S2 84	S1 69,3 S2 70.8	Pre-treated and naïve.	S1 40% S2 57%	2013–2016	10.6 months	^177^Lu-PSMA-617 and ^177^Lu-PSMA-I&T
Baum et al. [22]	2	Single-center prospective	Germany	S 56	72	Pre-treated	S 80.4%	2013–2015	Median 15 months, total 28 months	^177^Lu-PSMA
LuPSMA trial Hofman et al. [23]	2	Prospective single-arm	Australia	S 30	71	Pre-treated	S 97%	2015–2016	12 weeks	^177^Lu-PSMA-617
Violet et al. [24]	2	Prospective	Australia	S 50	71	Pre-treated	S 64%	2015–2016, 2017	Mean 31.4 months	^177^Lu-PSMA-617
Rahbar et al. [25]	-	Retrospective	Germany	S 104	70	Pre-treated	Any—67% decline ≥50% −33%	2014–2016	Overall survival	^177^Lu-PSMA-617
*Localized PC*										
Golan et al. [26]	1	Single-arm clinical trial	Israel	S 14	67	Primary	after 2 doses, 64% after 3 doses, 75%	2019–2021	12 months	^177^Lu-PSMA
LuTectomy Dhiantravan et al. [27]	1/2	Single-arm Study open-label nonrandomised	Australia	S 20	66	Primary	45%	2020–2022	36 months	^177^Lu-PSMA (later prostatectomy + pelvic nodes dissection)
*mHSPC*										
Privé et al. [28]	1	Prospective Pilot Study	Netherlands	S 10	67.2	Primary	No Data	2018–2019	10.6 months	^177^Lu-PSMA
BULLSEYE trial Prive et al. [29]	2	Two-arm randomized open-label	Netherlands	S 29 C 29	No data	only local therapy, no hormonal or chemotherapy	No data	2020–2023	6 months	^177^Lu-PSMA-I&T

**Table 2 cancers-16-02433-t002:** Information summary of 12 studies on the use of ^177^Lu-PSMA in the treatment of prostate cancer (original work).

Study	Inclusion Criteria	Exclusion Criteria	HR, OR	Adverse Events	Notable Findings	Overall Effect
*mCRPC*						
VISION trial, Sartor et al. [9]	Adults with CRPC with at least one metastatic lesion Disease progression after anti-androgen treatment (abiraterone and enzalutamide)	Any PSMA-negative lesions Eastern Cooperative Oncology Group performance score of—through 2 (0–5 scale) Life expectancy of at least 6 months Adequate organ/bone marrow function	HR for progression or death: 0.40 (CI 99.2%) HR for overall survival: 0.62 (CI 95%)	Fatigue Dry mouth Nausea Anemia Back pain Arthralgia Decreased appetite Constipation Diarrhea Vomiting Thrombocytopenia Lymphopenia Leukopenia	Low rate of adverse events Extended overall survival in a population with androgen-receptor pathway inhibitor-resistant disease	Overall survival versus standard care alone-: 4 months Progression-free versus standard care alone: 5.3 months
TheraP Hofman et al. [19]	PSMA-positive disease no sites of metastatic disease with discordant FDG-positive and PSMA-negative findings	Low uptake on [68Ga]Ga-PSMA-11 PET-CT discordant	HR for delayed progression: 0.63 (CI 95%)	Fatigue Pain Dry mouth Diarrhea Nausea Thrombocytopenia Dry eyes Anaemia Neuropathy Dysgeusia Haematuria Neutropenia Insomnia Vomiting Dizziness Leukopenia		Outcomes were better for cabazitaxel than for [¹⁷⁷Lu]Lu-PSMA-617
Satapathy et al. [20]	Metastatic disease on 68 Ga-PSMA-11 PET/CT with significant PSMA expression Chemotherapy-naive, however, patients with prior treatment of NAADs were included. ECOG 0-2 adequate hematological, renal, and liver function	Histological evidence of sarcomatous, spindle-cell or small-cell differentiation, and Sjogren syndrome	No data	Nausea/vomiting Constipation Fatigue Dryness of the mouth or eyes Abdominal pain Generalized pain Loss of weight or appetite Hematological toxicity Nephrotoxicity Hepatotoxicity	The outcomes are not significantly different between the two arms; one possible explanation for this could be the relatively higher tumor burden in the ^177^Lu-PSMA-617 arm. This is evident from the higher median baseline PSA.	^177^Lu-PSMA-617 was demonstrated to be non-inferior to docetaxel. Moreover, ^177^Lu-PSMA-617 was tolerated well vis-à-vis docetaxel, with less frequent grade 3/4 adverse events.
Barber et al. [21]	Selection for ^177^Lu-PRLT was based on PSMA-avid metastatic disease confirmed on pretherapy 68Ga-PSMA PET/CT imaging. In this study, patients were classified as either taxane che-motherapy pretreated (T-pretreated) or naïve (T-naïve) depending on whether they had received taxane-based chemotherapy (first- or second-line) prior to ^177^Lu-PRLT	T-naive patients who had been previously treated with non-taxane-based cytotoxic chemotherapy before ^177^Lu-PRLT were excluded from this analysis.	No data	Hematologic toxicity Anemia Leukocytopenia Thrombocytopenia Renal toxicity Creatinine Hepatic toxicity	^177^Lu-PRLT is a promising therapy in mCRPC, with encouraging outcomes and minimal associated toxicity seen in both our T-naive and heavily pretreated patient cohorts.	^177^Lu-PRLT was safe, with minimal adverse effects evident during follow-up in both T-pretreated and T-naive patients.
Baum et al. [22]	PSMA expression, progressive mCRPC, rising prostate-specific antigen (PSA) levels, 68Ga-PSMA PET/CT before therapy, renal function, hematologic status, previous treatments, and Karnofsky Performance Status score	No data	No data	Mild reversible xerostomia, reduction in erythrocyte and leukocyte counts, leukocytopenia	Safe and effective, objective responses with minimal toxicity in patients whose prostate cancer had progressed despite all standard treatments	PSMA RLT with ^177^Lu-PSMA is feasible, safe, and effective in end-stage progressive mCRPC with appropriate selection and follow-up of patients by 68Ga-PSMA PET/CT through application of the concept of theranostics.
LuPSMA trial Hofman et al. [23]	Pathologically (adenocarcinoma) confirmed metastatic castration-resistant prostate cancer with progressive disease after standard treatments, including taxane-based chemotherapy and second-generation anti-androgen treatment (abiraterone, enzalutamide, or both).	Low PSMA-avidity or FDG-discordant disease	No data	Dry mouth Lymphocytopenia Thrombocytopenia Fatigue Nausea Anaemia Neutropenia Pain Vomiting Anorexia Dry eyes Weight loss Disseminated intravascular coagulation Oculomotor nerve disorder Spinal fracture Hip fracture	Improved cognitive functioning and insomnia	^177^Lu-PSMA was well tolerated and could be a useful therapeutic option for mCRPC.
Violet et al. [24]	Pathologically confirmed mCRPC with progressive disease after taxane-based chemotherapy and second-generation antiandrogen therapy (abiraterone, enzalutamide, or both). Radiographic progression or new pain in an area of radiographically evident disease and an ECOG Performance Status of 2 or less must have occurred within 12 months. Tumor SUVmax (standardized uptake value) had to be at least 1.5 times the liver SUVmean to indicate PSMA intensity at disease sites.	Significant hematologic abnormalities, renal or liver insufficiency Prior to radiotherapy within 6 weeks, or uncontrolled disease. PET showed major discordant disease sites: 18F-FDG-positive and PSMA-negative.	No data	Dry mouth Lymphocytopenia Thrombocytopenia Fatigue Nausea Anemia Neutropenia Bone pain Vomiting Anorexia Dry eyes Renal injury	Long-term outcomes (after LuPSMA trial) high therapeutic efficacy and low toxicity, improvement in QoL in multiple domains. high response rates but less durable responses in patients rechallenged with ^177^Lu-PSMA on progression	High response rates, low toxicity, and improved QoL with ^177^Lu-PSMA radioligand therapy. On progression, rechallenge ^177^Lu-PSMA demonstrated higher response rates than other systemic therapies.
Rakhbar et al. [25]	Interdisciplinary tumor board, lacking other therapeutic options and disease progression despite established therapies according to mCRPC management guidelines, treated with ^177^Lu-PSMA-617, received at least one line of chemotherapy (docetaxel and/or cabazitaxel) and at least one next-generation antihormonal therapy (enzalutamide and/or abiraterone), and PSMA imaging was performed in all patients using 68Ga-PSMA-11 PET-CT or PET-MRI.	Bone marrow depression, Abnormal renal or liver function, upper urinary tract obstruction	HR 0.38 (CI 95%) (any PSA response vs. PSA progression) HR 0.28 (CI 95%) (PSA decline ≥20.87% as cutpoint)	No data	PSA decline ≥20.87% as the most noticeable cut-off prognosticating longer survival, which remained an independent prognosticator of improved OS in the multivariate analysis.	^177^Lu-PSMA-617 RLT is a new effective therapeutic and seems to prolong survival in patients with advanced mCRPC pretreated with chemotherapy, abiraterone, and/or enzalutamide.
*Localized PC*						
Golan et al. [26]	Adult with high-risk localized prostate cancer (cT3/4, Gleason score ≥ 8, prostate biopsy, PSA ≥ 20 ng/dL) or loco-regional prostate cancer (pelvic lymphadenopathy ≥ 2 cm on axial imaging). PET/CT PSMA showed higher PSMA expression than the liver. There were no PET FDG-positive sites without high PSMA expression. Patients should have an ECOG performance status score of 1 or lower. Life expectancy is >10 years.	Radiotherapy within two months. Combining nephrotoxic drugs. Distal lymphadenopathy, visceral, or bone metastases. Renal or liver insufficiency, hematologic abnormalities	No data	Nausea, Fatigue, Xerostomia	^177^Lu-PSMA, followed by RARP, seems to be tolerated similarly to RARP alone and has a high safety profile.	Absence of serious systemic adverse effects. Histological examination shows typical radiation-induced changes.
LuTectomy Dhiantravan et al. [27]	An adult with histologically verified adenocarcinoma of the prostate, planned for radical prostatectomy and pelvic lymph node dissection with the objective to cure. Classified as having high- or intermediate risk localized or loco-regional prostate cancer, in accordance with the European Association of Urology’s criteria. Significant PSMA avidity on 68 Ga-PSMA PET/CT—SUVmax of 20 or greater. Hematological and serum biochemistry parameters fall within normal ranges.	Rare prostate cancer with neuroendocrine or other pathology. Past prostate cancer treatment with radiotherapy or androgen deprivation. Any metastasis or lymph nodes above the common iliac artery bifurcation. Renal impairment Sjogren syndrome Any conditions, treatments, or laboratory abnormalities that could confound the trial results	No data	No data	No data	The first results will be known in June 2023.
*mHSPC*						
Privé et al. [28]	Prostate histological adenocarcinoma, Over 50 years old Prior local therapy with biochemical recurrence or clinical progression PSA doubling time: <6 months Local treatment is no longer viable 68Ga-PSMA-11-PET/CT positive metastases in bones and/or lymph nodes: ≥1 to 10 metastases (at least 1 lesion ≥1 cm for dosimetry studies) Free of visceral metastases. Normal liver function and blood count	68Ga-PSMA-11-PET/CT showed no lesions below the liver uptake. An alternative to prostate adenocarcinoma. Any medical condition that the investigator believes will affect patients’ clinical status in this trial. Prior hip replacement surgery Contraindications for MRI, Glucagon, or Buscopan	No data	Fatigue Nausea Xerostomia Rash Pain	Findings regarding both toxicity and efficacy are performed in a small cohort of selected patients.	^177^Lu-PSMA appeared to be a feasible and safe treatment modality in ten patients with low-volume mHSPC. Although the patients were treated with a relatively low dose of ^177^Lu-PSMA, the majority of patients showed a promising response to this therapy. This supports the need for the following trials to further evaluate the efficacy of ^177^Lu-PSMA in low-volume metastatic disease as well as in HSPC.
BULLSEYE trial Privé et al. [29]	Histologically proven prostate adenocarcinoma Biochemical recurrence and PSA-doubling time < 6 months. Positive 18F-PSMA-PET/CT metastases in bones and/or lymph nodes (N1/M1ab): ≥1 to 5 metastases. Local treatment is no longer possible. No prior hormonal, docetaxel or cabazitaxel therapy Lesion with an SUVmax > 15 ECOG Performance Status: 0–1. Life expectancy is >6 months. Normal liver and renal function, along with blood count	Known subtypes other than prostate adenocarcinoma, PSMA-based radioligand treatment, visceral or brain metastases, Any medical condition the investigator believes will affect patients’ clinical status in this trial. Prior hip replacement. Sjogren’s Syndrome Another active cancer is prostate cancer. Sexually active patients who cannot use medically acceptable barrier contraception.	No data	The study is not finished.	Patients in the SOC arm are eligible to receive ^177^Lu-PSMA-I&T after meeting the primary study objective, which is the fraction of patients who show disease progression during the study follow-up.	The study is not finished.

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
