# Peer review of "Role of Lutetium Radioligand Therapy in Prostate Cancer"

_cancers, 2024, doi:10.3390/cancers16132433_

Round 1

Reviewer 1 Report

Comments and Suggestions for Authors

The manuscript summarized current literature on the utilization of lutetium radio-ligands in different clinical settings. Here are the comments 

1. Table 2 - first row - please spell out "exclusion criteria" in completion. The description of studies needs to be in a uniformed format - this is especially notable for the mHSPC studies. All the mHSPC studies the including criteria should be more concise with summary instead of directly coping from the original study.  Also what dose the abbreviation "ND" mean?

2. Page 26 - the entire conclusion, author contributions, funding and conflits of interest part is missing

3. The numbering of the sub-titles are very confusion. For example, "mCRPC" was numbered 2.1 (line 75) but "localised" was numbered 2.2.1 (line 161). They should be on the same level. 

4. As mentioned already, the manuscript will need careful proof reading/polish by a native reader to improve readability.

Comments on the Quality of English Language

The manuscript needs proof reading by a native English speaker. 

Author Response

The manuscript summarized current literature on the utilization of lutetium radio-ligands in different clinical settings. Here are the comments

Response: We would like to thank the Reviewer for their thorough review of our manuscript, and valuable comments which improved the quality of our manuscript.

  1. Table 2 - first row - please spell out "exclusion criteria" in completion. The description of studies needs to be in a uniformed format - this is especially notable for the mHSPC studies. All the mHSPC studies the including criteria should be more concise with summary instead of directly coping from the original study. Also what dose the abbreviation "ND" mean?

Response: We thank the Reviewer for this comment. We have made the suggested changes, including spelling of “exclusion criteria”, unifying the reporting of study descriptions, and paraphrasing the inclusion criteria of the mHSPC studies to be more concise.

The abbreviation “ND” stands for No data; we apologize for the confusion, and we have clarified this in the manuscript.

  1. Page 26 - the entire conclusion, author contributions, funding and conflits of interest part is missing

Response: We apologize to the reviewer. We have added these information to the manuscript.

  1. The numbering of the sub-titles are very confusion. For example, "mCRPC" was numbered 2.1 (line 75) but "localised" was numbered 2.2.1 (line 161). They should be on the same level.

Response: We agree with the reviewer that in its current state, the numbering could be regarded confusing. We have simplified the numbering (2, 3, etc.) throughout the manuscript.

  1. As mentioned already, the manuscript will need careful proof reading/polish by a native reader to improve readability.

Response: We apologize to the reviewer for the language errors in the manuscript. We performed proofreading of the manuscript to improve the quality of English, and hope that the Reviewer will find it satisfactory.

Reviewer 2 Report

Comments and Suggestions for Authors

The topic is interesting and relevant to the journal.

However, the unclear organization through out the manuscript is found in every section. For example, 1.2 Pharmacodynamics and pharmacokinetics can be included in the introduction. section 2, mCRPC, Localised, and mHSPC should be list 2.1, 2.2 and 2.3, and in the end, no conclusion in the manuscript. 

All case studies should be listed in the reference number in the table. the details of case studies need to be discussed in the main text of the paper. 

Author Response

The topic is interesting and relevant to the journal.

However, the unclear organization through out the manuscript is found in every section. For example, 1.2 Pharmacodynamics and pharmacokinetics can be included in the introduction. section 2, mCRPC, Localised, and mHSPC should be list 2.1, 2.2 and 2.3, and in the end, no conclusion in the manuscript.

Response: We thank the Reviewer for their thorough review of our manuscript and constructive comments. We agree that the chapter organization could be regarded as confusing, and we have simplified it by numbering the sections as 2. mCRPS, 3. Localized, etc., throughout the manuscript. Additionally, as suggested by the Reviewer, we incorporated the pharmacodynamics and pharmacokinetics sections to the “Introduction”.

All case studies should be listed in the reference number in the table. the details of case studies need to be discussed in the main text of the paper.

Response: We thank the Reviewer for their comment; however, we are not sure if we understood the comment correctly. We have made sure that the case reports referenced in the manuscript are discussed in the text, and referenced in the tables summarising the available literature. We would appreciate the Reviewer’s feedback whether our corrections are satisfactory.

Reviewer 3 Report

Comments and Suggestions for Authors

The article discusses the advances in prostate cancer treatment, focusing on the use of Lutetium-177-PSMA (Lu-PSMA) radioligand therapy. This novel treatment targets prostate-specific membrane antigen (PSMA) to deliver radiation directly to cancer cells, minimizing damage to healthy tissues. Lu-PSMA therapy has shown promising results in managing metastatic castration-resistant prostate cancer (mCRPC) and improving patients' quality of life with manageable side effects. Emerging data suggest its efficacy and safety, though further research is needed to validate these findings in larger patient cohorts.

  1. Limited Long-term Data: The article highlights the lack of long-term adverse effect data, which is crucial for understanding the full impact of Lu-PSMA therapy.
  2. Control Arm Issues: The VISION trial's control arm had sub-standard treatment methods, possibly skewing the comparative efficacy results of Lu-PSMA.
  3. Underreporting of Adverse Events: Retrospective analyses might underreport adverse effects, emphasizing the need for more prospective trials for accurate monitoring.
  4. Generalizability: The promising results from small-scale studies and heavily pretreated patient groups need validation through larger, more diverse cohorts to confirm the generalizability of Lu-PSMA therapy's benefits.

Author Response

The article discusses the advances in prostate cancer treatment, focusing on the use of Lutetium-177-PSMA (Lu-PSMA) radioligand therapy. This novel treatment targets prostate-specific membrane antigen (PSMA) to deliver radiation directly to cancer cells, minimizing damage to healthy tissues. Lu-PSMA therapy has shown promising results in managing metastatic castration-resistant prostate cancer (mCRPC) and improving patients' quality of life with manageable side effects. Emerging data suggest its efficacy and safety, though further research is needed to validate these findings in larger patient cohorts.

Limited Long-term Data: The article highlights the lack of long-term adverse effect data, which is crucial for understanding the full impact of Lu-PSMA therapy.

Control Arm Issues: The VISION trial's control arm had sub-standard treatment methods, possibly skewing the comparative efficacy results of Lu-PSMA.

Underreporting of Adverse Events: Retrospective analyses might underreport adverse effects, emphasizing the need for more prospective trials for accurate monitoring.

Generalizability: The promising results from small-scale studies and heavily pretreated patient groups need validation through larger, more diverse cohorts to confirm the generalizability of Lu-PSMA therapy's benefits.

Response: We thank the Reviewer for their comments and positive appraisal of our manuscript’s quality. We agree with the Reviewer’s comments and acknowledge that the mentioned aspects of Lu-PSMA therapy are discussed in our manuscript.

Round 2

Reviewer 2 Report

Comments and Suggestions for Authors

After extensively revised, the manuscript is now ready to be accepted.